# Hollow Microcavity Electrode for Enhancing Light Extraction

**DOI:** 10.3390/mi15030328

**Published:** 2024-02-27

**Authors:** Seonghyeon Park, Byeongwoo Kang, Seungwon Lee, Jian Cheng Bi, Jaewon Park, Young Hyun Hwang, Jun-Young Park, Ha Hwang, Young Wook Park, Byeong-Kwon Ju

**Affiliations:** 1Display and Nanosensor Laboratory, Department of Electrical Engineering, Korea University, Seoul 02841, Republic of Korea; gohnd@korea.ac.kr (S.P.); kang7369@korea.ac.kr (B.K.); lswon96@korea.ac.kr (S.L.); vlfrkatjd@korea.ac.kr (J.C.B.); pkjaewon@korea.ac.kr (J.P.); aksyp@korea.ac.kr (Y.H.H.); mrjoon123@korea.ac.kr (J.-Y.P.); ha08.hwang@samsung.com (H.H.); 2Department of Semiconductor and Display Engineering, Sun Moon University, Asan 31460, Republic of Korea

**Keywords:** laser interference lithography, periodic array, hollow structure, nanoscale vacuum photonic crystal layer, finite-difference time-domain simulation, microcavity, OLED

## Abstract

Luminous efficiency is a pivotal factor for assessing the performance of optoelectronic devices, wherein light loss caused by diverse factors is harvested and converted into the radiative mode. In this study, we demonstrate a nanoscale vacuum photonic crystal layer (nVPCL) for light extraction enhancement. A corrugated semi-transparent electrode incorporating a periodic hollow-structure array was designed through a simulation that utilizes finite-difference time-domain computational analysis. The corrugated profile, stemming from the periodic hollow structure, was fabricated using laser interference lithography, which allows the precise engineering of various geometrical parameters by controlling the process conditions. The semi-transparent electrode consisted of a 15 nm thick Ag film, which acted as the exit mirror and induced microcavity resonance. When applied to a conventional green organic light-emitting diode (OLED) structure, the optimized nVPCL-integrated device demonstrated a 21.5% enhancement in external quantum efficiency compared to the reference device. Further, the full width at half maximum exhibited a 27.5% reduction compared to that of the reference device, demonstrating improved color purity. This study presents a novel approach by applying a hybrid thin film electrode design to optoelectronic devices to enhance optical efficiency and color purity.

## 1. Introduction

Optoelectronic devices play a significant role in the fields of optics and electronics because of their ability to convert electrical energy into light energy and vice versa. The luminous efficiency and color purity of emitted or absorbed light are critical factors that can affect the performance of these devices [1,2,3,4]. Representative luminescent devices, such as organic light-emitting diodes (OLEDs), require high luminous efficiency to reduce power consumption and high color purity to display a broader spectrum of colors [5,6]. However, most of the light generated by the devices cannot escape but is internally extinguished because of various factors [7,8,9].

The external quantum efficiency (EQE) serves as a measure of the optical efficiency in OLEDs, signifying the quantity of photons emitted per injected carrier. The EQE is calculated by multiplying the internal quantum efficiency (IQE), number of photons generated within the device per injected carrier, and light extraction efficiency [10]. For fluorescent materials that utilize only a singlet exciton, the IQE is 25% [11,12]. However, materials employing both singlet and triplet excitons, such as phosphorescent and thermally activated delayed fluorescent materials, have the potential to achieve 100% efficiency [13,14,15]. Although an IQE of 100% can be theoretically achieved, the ratio of photons emitted into the air is approximately 20% of the number of photons generated. This photon loss occurs because of interactions between various materials in the device [16,17].

Many optoelectronic devices consist of multiple layers of thin films made of different materials such as organic, metal, and polymer materials [18,19,20]. OLEDs consist of multiple layers of organic materials and electrodes, which result in significant optical losses. As the photons exit the OLEDs, they undergo total internal reflection because of the refractive index mismatch among the components, including the organic layers and electrodes within the devices. Depending on their location, this phenomenon is referred to as the substrate or waveguide mode [21,22]. OLEDs suffer from energy losses due to the surface plasmon polariton (SPP) mode resulting from the interaction between photons and electromagnetic waves at the interface between the metal electrode and organic layers [23,24,25]. The outcoupling efficiency indicates the ability to release photons without being lost in various modes. Various methods can be applied to enhance the outward emission of the generated photons, including refractive index matching to prevent internal reflection and incorporating nanostructures within the device to manipulate the path of light [26,27,28,29,30].

The microcavity effect is an effective technique for improving the efficiency and color purity of a device by leveraging the interference of light. Light is reflected between the thick metal layer and semi-transparent reflective layer within the device, thereby creating a resonance frequency, such as the intrinsic vibration frequency of the luminescent material, to induce constructive interference. Precise design considerations of the thickness and refractive index of the device components are necessary for obtaining the desired resonance frequency [31,32,33]. 

In this study, we propose a hollow electrode with a thin film design to enhance light extraction through nanoscale vacuum photonic crystal layer (nVPCL) insertion (Figure 1). The nVPCL was fabricated using laser interference lithography (LIL), which offers large-area patterning and mask-free fabrication. We designed a periodic line structure and utilized the grating vector (k_g_) generated in the structure to guide the light within the device to exit externally through a shift in the lossy in-plane wave vector. The amplitude of the grating vector was controlled by adjusting the spacing of the periodic structure, thereby allowing a custom design for specific targeting wavelengths [34,35,36]. The nVPCL can be applied to optoelectronic devices for extracting photons trapped within the devices in the waveguide and SPP modes. Further, the vacuum component of the nVPCL, which has the refractive index of air, maximizes the refractive index difference from the electrode to guide and extract the internally reflected light between the substrate and the electrode [37,38]. We demonstrated the light extraction enhancement of the nVPCL by applying it to OLEDs, which suffer from lossy waveguide and SPP modes. Employing this structure as a semi-transparent reflective electrode can induce a microcavity effect, which can lead to improvements in efficiency and color purity [31]. We applied a Ag thin film as the electrode of the nVPCL to induce a microcavity and used indium zinc oxide (IZO), which does not require heat treatment and can alleviate electrode roughness [39]. The nVPCL structure was optimized through the finite-difference time-domain (FDTD) simulation method to extract light in the green wavelength range. 

## 2. Materials and Methods

### 2.1. Laser Interference Lithography

The LIL process is a nanoscale patterning technique employed for fabricating periodic nanostructures. This process involves the overlay of two or more coherent light beams using an interferometer. The laser employed in this study was a Gaussian beam derived from a frequency-doubled Ar-ion laser, featuring a wavelength of 257 nm. The laser beam passed through the objective lens and pinhole, thereby producing a divergent beam incident on the sample stage. A Llyod’s mirror was positioned on the stage to adjust the periodicity of the interference pattern exposed to the sample by controlling the incident angle. A single laser exposure produced a one-dimensional line pattern using Lloyd’s interferometer. Additional exposure is required with a 90° rotation of the sample to attain a two-dimensional pattern with square symmetry, such as a hole or dot pattern. Adjusting various factors such as the wavelength of the laser, lens configuration, distance, and pinhole size allows for the design of an exposure area tailored to the size of the sample, enabling application to samples of various sizes. There are many methods, such as DTL (Displacement Talbot Lithography), that utilize light interference to form nanosize patterns [40,41]. However, unlike those methods, LIL does not require a mask, allowing for easy modification of the structure design by adjusting the angle of Lloyd’s mirror and development time. Furthermore, it has a relatively simple system, allowing for patterns to be formed quickly and easily.

### 2.2. Fabrication of a Nanoscale Vacuum Photonic Crystal Layer

Figure 2 shows the fabrication process for the hollow microcavity electrodes. The nVPCL was produced on a glass substrate (Eagle XG, Corning Inc., Corning, NY, USA). The glass substrate was cleaned through sequential ultrasonication in acetone, methanol, and deionized water for 15 min each. A negative laser photoresist (AR-N4240, Allresist, Strausberg, Germany) was then applied through spin-coating at 6000 rpm for 40 s and subjected to a soft baking process at 105 °C for 60 s on a hot plate. The desired thickness was achieved by diluting the photoresist with a thinner solution (AR 300–12, Allresist, Strausberg, Germany) in a 1:5 ratio. This approach ensured the fabrication of a structure with an appropriate height, maintaining stable electrical properties without adopting any additional layers, such as a smoothing layer. The sample was subsequently structured using a LIL system, selected for its benefits in large-area patterning. The sample was exposed to a frequency-doubled Ar-ion laser at an energy of 12 mJ∙cm^−2^, resulting in the formation of periodic nanoscale lines, as shown in Figure 3. A schematic of the LIL process and laser equipment system used in this study is also presented in Figure 3. The periodicity of the laser interference pattern can be engineered by controlling the stage rotation. After post-exposure baking on the hot plate at 105 °C for 90 s, the exposed photoresist underwent development using a developer (AZ 300 MIF, Merck, Darmstadt, Germany), thereby resulting in the formation of a periodic line photoresist pattern. Subsequently, a semi-transparent electrode was fabricated using a vacuum equipment. A thermal evaporator (Korea Vacuum Tech, Ltd., Gimpo, Republic of Korea) was used under high-vacuum (10^−7^ Torr) conditions to deposit 15 nm of the thin film Ag layer at a rate of 0.2 Å/s. The thickness was monitored by the QCM (Quartz Crystal Microbalance) sensor within the equipment. Subsequently, 105 nm of IZO was deposited using radiofrequency (RF) sputtering (Korea Vacuum Tech, Ltd., Gimpo, Republic of Korea) in the 4.0 × 10^−6^ Torr vacuum chamber to reinforce the stability of the charge injection. The applied RF power was 150 W and the Ar gas flow rate was 5.8 sccm. Under these conditions, the deposition rate was experimentally measured and the time required to achieve the desired thickness was determined. After the process was completed, the thickness of the deposited layer was validated using field emission scanning electron microscopy (FE-SEM, S-4800, Hitachi, Ltd., Tokyo, Japan). The photoresist pattern was eliminated entirely using a photoresist remover (AR 300-76, Allresist, Strausberg, Germany) to create a hollow structure underneath. This hollow structure acted as the periodic photonic crystal structure and was therefore referred to as the nVPCL. The nVPCL was fabricated between the glass substrate and Ag/IZO composite electrode by removing the line-pattern photoresist. The combination of the nVPCL and ultrathin Ag and IZO layers on top formed a corrugated semi-transparent electrode for enhancing light extraction. 

Figure 3 illustrates the three adjustable parameters of the process: pitch, width, and height. The pitch of the pattern can be defined by
(1)$$Pitch(\Lambda )=\frac{\lambda_{laser}}{2sin\theta }$$

The width was determined based on the development time during the development process and the height was defined based on the spin-coating speed in revolutions per minute. A vacuum structure could not be established when processing the photoresist removal with only the stacked Ag film because of the low thickness of Ag. Consequently, as shown in Appendix A, the nVPCL was not defined and the Ag film failed to form a corrugated profile and collapsed. Therefore, photoresist removal was performed after Ag and IZO deposition to produce the nVPCL.

### 2.3. Nanophotonic Computational Analysis

The hollow microcavity electrode was analyzed computationally using the FDTD software (Release: 2021 R1.2, Version: 8.25.2621) (ANSYS, Inc., Canonsburg, PA, USA). The emission source was an oscillating dipole and the distribution of the electromagnetic waves was calculated using the FDTD method with Maxwell’s equations. The nanostructure was divided into mesh units called Yee’s cells and the distributions of electric and magnetic fields were computed at the mesh boundaries. The simulation accurately captured the dimension and refractive indices of the layered structure. At a wavelength of 525 nm, the refractive indices of each material were as follows: glass substrate (1.53), Ag (0.13), IZO (2.01), NPB (1.81), Alq_3_ (1.72), and Al (0.84). A thin film analyzer (F-20, Filmetrics, Inc., San Diego, CA, USA) was employed to ascertain the refractive indices of individual layers. Diverse materials forming the device were applied to a flat silicon wafer using a QCM sensor in the thermal evaporator to achieve a thickness of precisely 100 nm and the reflected signal was analyzed to determine the refractive index and extinction coefficient of each material. These measurement values were applied to the simulations.

In addition, to alleviate undesired light interference caused by edge reflections, the simulation boundaries were configured as perfectly matched layers (PML), designating only the cathode side as the metal boundary for absorption. The *x*-axis width was set to exceed 20 μm to accurately represent lateral light propagation characteristics. The simulation employed 838,400 meshes (3200 × 262). In addition, in the case of the OLED, the result from three distinct dipole orientations (*x*-, *y*-, and *z*-polarized) were averaged to accommodate the isotropic emitter characteristics. The evaluation of the light extraction enhancement factor involved a comparison between the measured integrated light intensity from the far-field monitor and a reference.

### 2.4. Device Fabrication

After preparing the substrate/electrode sample, the positive photoresist (AZ GXR-601, Merck, Darmstadt, Germany) was spin-coated for 30 s at 3000 rpm, followed by annealing at 105 °C for 60 s on a hot plate for soft baking. Photolithography was conducted using a chrome mask designed to emit pixels of 2 × 2 mm^2^ and this process was performed using a mask aligner (CA-6M, Shinumst Co., Ltd., Daejeon, Republic of Korea). After photolithography, the exposed photoresist underwent processing using the developer (AZ 300 MIF, Merck, Darmstadt, Germany), which resulted in the formation of active areas in a 2 × 2 array insulated from each other. For the surface treatment, UV ozone treatment was conducted using a UV cleaning system (AH-1700, AHTECH LTS Co., Ltd., Anyang, Republic of Korea) for 2 min and 40 s, whereas oxygen plasma treatment was carried out using a vacuum plasma system (CUTE, Femto Science Inc., Hwaseong, Republic of Korea) at 80 W and 15 sccm for 2 min and 40 s. Subsequently, the organic layer of the green OLED and metal layer were deposited using a thermal evaporator (Korea Vacuum Tech, Ltd., Gimpo, Republic of Korea) under high-vacuum (10^−7^ Torr) conditions. The structure of the green fluorescent OLED was as follows: 60 nm N,N′-bis(naphthalen-1-yl)-N,N′-bis(phenyl)-benzidine (NPB)/80 nm Tris(8-hydroxyquinolinato) aluminum (Alq3)/2 nm lithium fluoride (LiF)/100 nm aluminum (Al).

### 2.5. Measurements

The surface morphology of the nVPCL-integrated electrode was analyzed by FE-SEM and atomic force microscopy (AFM, XE-100, Park Systems Corp., Suwon, Republic of Korea). The measurement conditions for AFM included a scan rate of 1 Hz and a scan area of 10 × 10 μm^2^. The transmittance, reflectance, and haze of the nVPCL-integrated electrode were analyzed using an ultraviolet–visible–near infrared spectrophotometer (Cary 5000, Agilent Technologies, Santa Clara, CA, USA). The electroluminescence characteristics of the OLEDs were measured using a spectroradiometer (PR-670, JADAK, Syracuse, NY, USA) with a high-voltage source measurement unit (Keithley 237, Keithley Instruments, Inc., Cleveland, OH, USA) in a black box (Appendix A). 

## 3. Results and Discussion

### 3.1. Design and Fabrication of the Hollow Microcavity Electrode

The electrode was designed as an integrated structure consisting of a semi-transparent Ag thin film layer and an IZO layer to induce microcavity resonance. An ultrathin layer of Ag was deposited via thermal evaporation under high-vacuum conditions of 10^−7^ Torr. This layer served as an exit mirror, providing suitable reflectivity within the Fabry−Perot resonator model [42,43]. The IZO layer, applied with adequate thickness, smoothed the surface roughness of the thin Ag layer, thereby suppressing undesired scattering phenomena or current leakage. Utilizing only a thin Ag layer as an electrode not only results in insufficient charge injection but also introduces challenges such as relatively high surface roughness, thereby leading to issues such as dark spots or non-uniform operational characteristics in the electrical device because of localized electric field enhancement [44,45]. Therefore, a layer of IZO was added on the Ag thin film to function as a buffer to mitigate the instability arising from surface roughness while simultaneously acting as an auxiliary electrode to ensure enhanced conductivity.

Furthermore, the integrated Ag/IZO electrode was configured with a periodic corrugation profile to optimize light extraction at a specific wavelength. In numerous optoelectrical devices, most of the light generated inside the device is not emitted; instead, it is absorbed and lost within internal structures. In OLEDs, in addition to the light lost because of total reflection caused by the highly refractive internal constituent layers, the SPP mode is induced through the near-field interaction of the oscillating dipole. Given the proximity of the emissive dipole to the cathode, which is only a nanometer long, electrically generated excitons can couple with SPPs on the cathode surface through near-field energy transfer. 

The dispersion relationship of the SPP mode exhibits behavior similar to that of photons; however, it ultimately saturates to the surface plasmon frequency. As shown in Figure 4a, the in-plane wave vector of the SPP mode does not overlap with the air mode cone, and therefore, it cannot be captured in the air. Instead, they are absorbed and lost at the surface of the metal cathode. Consequently, in conventional OLED devices, SPP modes are characterized as nonradiative, lossy modes, thereby leading to energy coupling to surface plasmons and resulting in 40% energy loss compared to the initially generated photons.

As shown in Figure 4, in instances where a periodic structure that is coplanar with the lost light is formed, the phenomenon of light extraction occurs according to the Bragg diffraction principle. The diffraction of light within the visible spectrum is achievable when the periodic structure is at the nanoscale, thereby resulting in the conversion of trapped light into the air mode. The trapped light caused by total internal reflection, known as the waveguide mode, and the light absorbed because of the SPP-coupled energy (SPP mode) can be characterized as the in-plane wavevector component (k_x_). Wave vectors attributed to the waveguide and SPP modes are denoted as k_WG_ and k_SPP_, respectively. A grating vector (kg) was formed within the same plane by implementing a nanoscale periodic structure. Therefore, the in-plane wave vectors can shift within the air cone because of the grating vector component, thereby leading to the extraction of the SPP mode, as shown in Figure 4a. Here, ko represents the wave vector of light in vacuum, where n, k′_x_, and θ represent the diffraction order, shifted in-plane wavevector component, and diffraction angle of the scattered light in free space, respectively.

Bragg diffraction can be expressed as
(2)$$k_{o}=\frac{2\pi }{\lambda },k_{g}=\frac{2\pi }{\Lambda }.$$
(3)$${k'}_{x}=k_{o}sin\theta =k_{x}\pm n∙k_{g}.$$

Figure 4b shows that the SPP mode can be converted into the air mode depending on the amplitude of the grating vector and the diffraction order. Figure 4c shows the wave vector system in the hollow microcavity electrode. The grating vector is formed in the in-plane direction by the periodic dimension of the nVPCL, which leads to the diffraction of the waveguide and SPP modes. The wavelength of the light is significantly influenced by the amplitude of the grating vector, which is defined by the periodicity of the nanostructure. Therefore, the light of a desired wavelength can be extracted via the meticulous design of the geometric parameters of the nanostructure. In the hollow microcavity electrode, the SPP mode in the thin Ag film can be converted into the air mode, and the waveguide mode within the glass substrate and IZO layer can be extracted. The in-plane wave vector (k_x_) can be represented by either k_WG_ or k_SPP_. 

### 3.2. FDTD Simulation for Bragg Diffraction and the Microcavity Effect

The periodicity of nanostructures with patterns was customized based on the targeted wavelength, depending on the optoelectronic device. The structure of the nVPCL was optimized using the FDTD simulation software, targeting a green wavelength of 525 nm. The simulation was performed by employing an OLED structure that lost a significant amount of light because of the waveguide and SPP modes in the optoelectronic devices. Figure 5 shows the analysis model used in the simulation. Figure 5a shows the conventional structure of a green fluorescent OLED [46] and Figure 5b,c show the addition of a Ag thin film to induce the microcavity effect, demonstrating the planar-cavity structure without the nVPCL and the corrugated-cavity structure with the nVPCL, respectively. In the planar structure, source dipoles were oriented in the *x*-, *y*-, and *z*-directions and the final distribution was achieved by averaging the results from the three orientations. This approach was adopted to capture the isotropic emission characteristics of the actual device. 

Considering the overall thickness of the OLED structure is essential for optimizing the pitch, width, and height of the nVPCL and simultaneously inducing the microcavity effect. Therefore, the thicknesses of the Ag thin film and IZO must be determined. The thickness of the Ag thin film used as an exit mirror can significantly influence the transmittance because it determines the intensity of the electric field (E-field) [47]. Therefore, it should be designed to have an appropriate thickness. The thickness of the IZO, which is used as a component of the anode, influences the cavity length. We used a conventional OLED structure without altering the thickness of the organic layer in the OLED and the resonance condition was determined by adjusting only the thickness of the IZO. Considering these factors, simulations were conducted to determine the appropriate thicknesses of the Ag thin films and IZO (Figure 6). In Appendix A, the contour plot indicated that the peak wavelength had negligible dependence on the Ag and IZO thicknesses. The optimal thickness of Ag providing maximum enhancement was found to be 15 nm, considering the reflection and transmittance for the target green wavelength of 525 nm. The IZO thickness, at which the enhancement peak appeared at 525 nm, was determined to be 105 nm by adjusting the cavity length to determine the enhancement wavelength. 

We applied the optimized Ag thin film and IZO thicknesses to a conventional green fluorescent OLED and swept the pitch, width, and height of the nVPCL structure, targeting a wavelength of 525 nm. Figure 7a presents a contour plot from the simulations aimed at identifying the pitch of the nVPCL structure maximizing the enhancement at 525 nm; Figure 7b graphically illustrates the light efficiency enhancement factor according to the pitch. The overall thickness of the OLED was designed to be the cavity length for the 525 nm resonance, which resulted in values higher than those of the conventional reference device for all pitch ranges, with a maximum enhancement peak observed at a pitch of 700 nm. Figure 7c shows the optimal width of the nVPCL structure. We defined the ratio of width to pitch as the duty cycle (dc), which can be expressed as
(4)$$Duty\;cycle\left(dc\right)=\frac{Width\;of\;the\;nVPCL}{Pitch\;of\;the\;nVPCL}\times 100\;[\%]$$

The proportion of the nVPCL in the entire area increased with an increase in the duty cycle, thereby leading to an increase in the enhancement factor. Maximum enhancement was observed at a duty cycle of 70, after which it exhibited a decreasing trend. Figure 7d shows the optimal height of the nVPCL structure. The height showed a maximum enhancement at 45 nm, followed by a subsequent decrease. Based on the results shown in Figure 7, we determined the optimal design of the nVPCL with a pitch, width, and height of 700, 480, and 45 nm, respectively, for green light enhancement.

We applied the device structure under the conditions described above and simulated the behavior of the E-field intensity across the entire device to confirm the extraction of the SPP mode (Figure 8). The simulation was conducted under the condition of transverse magnetic (TM) polarized dipoles because the presence of the SPP mode was limited to transverse magnetic (TM) polarization [48]. Figure 8a shows that the light in the planar-cavity device without the nVPCL structure was trapped and could not escape to the air mode at the Al/organic and Ag/glass interfaces. As shown in Figure 8b, the nVPCL structure facilitated the extraction of light into the air mode, thereby reducing the E-field at each interface. 

Figure 9 shows the integrated field intensity to quantitatively represent the results shown in Figure 8. A decrease in the E-field at the Al/organic and Ag/glass interfaces compared to that of the planar-cavity device was observed in the corrugated-cavity device with the nVPCL structure. In addition, the E-field intensity in the air mode increased in the corrugated-cavity device, thereby suggesting that the light trapped inside the component was effectively extracted outward because of the presence of the nVPCL. The E-field intensity at the IZO/Ag interface exhibited a slight increase attributed to the interference of the periodic structure where the nVPCL was inserted. However, there was an overall increase in the E-field intensity in the air mode. This suggests that the energy dissipated within the component by the waveguide and SPP modes could be effectively converted into other modes.

### 3.3. Analysis of the nVPCL-Integrated Electrode

An nVPCL was fabricated using the optimized results obtained from the simulations. During the LIL process, the laser incident angle was adjusted to 10.58° through stage rotation to fabricate an optimized pitch for the 525 nm wavelength in the nVPCL structure. Various measurements were conducted to verify the feasibility of the nVPCL-integrated electrodes. Figure 10 shows the SEM image and surface morphology of the nVPCL-integrated electrode with different views. Figure 10a illustrates the periodic line patterning achieved using the photoresist via the LIL process. We formed a thin photoresist line pattern of the desired size. Figure 10b shows the appearance after the deposition of Ag and IZO. Figure 10c shows the effective removal of the residual photoresist through the cleaning process using the remover, which resulted in well-formed vacuum components. As shown in Figure 10c, the nVPCL was created below the Ag/IZO anode and the overall structure functioned as a hollow microcavity electrode. 

The surface stability and uniformity of the fabricated nVPCL were investigated to assess its suitability as an electrode material for practical optoelectronic devices. The AFM images were analyzed to confirm the smoothing effect of the IZO layer. The electrode without the IZO layer was analyzed without removing the photoresist, which revealed no difference in the smoothing effect. Figure 11 shows the top-view AFM images and surface morphology of the nVPCL-integrated structures. Figure 11a,b show the corrugated Ag and Ag/IZO structures with IZO deposited on the structure shown in (a). The line patterning was well-maintained even when Ag and IZO were deposited on the line-patterned structure using the photoresist. 

Table 1 presents the roughness values of the different structures measured using AFM. Additionally, Appendix A shows the AFM images of the structures of Ag and Ag/IZO. We measured the root-mean-square deviation (Rq) and arithmetical mean height (Ra) of each electrode to assess the roughness and uniformity of the electrode surface. The decrease in roughness at Rq and Ra observed upon the addition of the IZO layer suggested that the IZO layer contributed to the surface smoothing effects. Furthermore, we measured the sheet resistance to understand the effect of the roughness of the nVPCL-integrated electrode on conductivity. The sheet resistance of the electrode in the Ag/IZO structure was measured to be 2.361 Ω/sq, whereas the sheet resistance of the electrode in the nVPCL/Ag/IZO structure was measured to be 2.588 Ω/sq (Appendix A). Despite the increase in surface roughness caused by the insertion of the nVPCL into the Ag/IZO structure, the sheet resistances of the two structures showed similar values. Therefore, the nVPCL-integrated electrode has a sufficiently low roughness and high conductivity, thereby making it suitable for use as an electrode in optoelectronic devices.

Figure 12 shows photographs of the fabricated nVPCL-integrated electrode and the Ag/IZO structure without the nVPCL. Both the Ag/IZO structure and the structure with the inserted nVPCL exhibited semi-transparent characteristics when observed visually. Further, the incorporation of the nVPCL resulted in slightly enhanced transparency. We measured the transmittance, reflectance, and haze to assess its suitability as an electrode capable of inducing a microcavity effect in optoelectronic devices (Appendix A). The specular transmittance of the nVPCL-integrated electrode was measured to be 37.99% at a wavelength of 525 nm, which was higher than the 26.84% observed for the Ag/IZO structure. This demonstrates that the periodic pattern of the nVPCL, along with a refractive index of 1, can efficiently prevent and extract light, thereby enhancing transmittance [49,50]. The reflectance at 525 nm was 43.86%, which was sufficient to induce a microcavity effect [51,52]. The haze at the same wavelength was measured to be 6.21%, which indicated a negligible scattering effect when incorporated with optoelectronic devices emitting light. 

### 3.4. Device Application

We applied the fabricated nVPCL-integrated electrode to OLEDs, which often suffer from significant optical losses because of the waveguide and SPP modes in optoelectronic devices. The OLED devices used a conventional fluorescent green OLED structure, which is consistent with the structure simulated earlier. We investigated the light extraction enhancement targeting a wavelength of 525 nm by incorporating the nVPCL. Figure 13 illustrates the EL characteristics of nVPCL-OLEDs. The device using an electrode with the nVPCL insertion was referred to as the corrugated-cavity device, whereas the device using an electrode with only an Ag thin film to induce the microcavity effect without the nVPCL was referred to as the planar-cavity device. As shown in Figure 13a, the current densities in the planar-cavity and corrugated-cavity devices with the addition of the Ag thin film increased compared to that of the reference device using only IZO electrodes. This increase was attributed to the ease of current injection with the addition of the Ag layer. The resonance phenomenon utilized in both the devices and the microcavity effect led to a higher emission of light at a wavelength of 525 nm. Consequently, the luminance increased compared to that of the reference device. In the corrugated-cavity device, the periodic patterning of the nVPCL along with a refractive index of 1 allowed for the additional outward extraction of light. Consequently, the luminance increased further, demonstrating the enhanced effectiveness of the nVPCL-OLEDs. Figure 13b illustrates the enhanced external quantum efficiency (EQE) of the corrugated-cavity device with respect to current density. It exhibited a 9.3% and 17.6% improvement compared to the planar-cavity and reference devices at a current density of 300 mA/cm^2^. Figure 13c shows the improvement in the EQE based on luminance. At 10,000 cd/m^2^, it exhibited a 9.4% enhancement over the planar-cavity device and a 21.5% improvement over the reference device. The efficiency of the reference device decreased sharply to 10,000 cd/m^2^, whereas the planar- and corrugated-cavity devices exhibited normal luminescence trends. This indicates that the two devices with the added Ag thin film operate more consistently at higher voltages compared to the devices that use conventional electrodes. The current efficiency of the corrugated-cavity device increased by 34.6% compared to that of the reference device at 10,000 cd/m^2^ (Appendix A). 

The EL spectra of the devices were recorded at a current density of 300 mA/cm^2^. Figure 14a shows that the emission spectra became sharper in both the planar-cavity and corrugated-cavity devices, inducing a microcavity effect, compared to that of the reference device. The full width at half maximum (FWHM) was 102 nm for the reference device, whereas it decreased by 33.3% to 68 nm for the planar-cavity device and by 27.5% to 74 nm for the corrugated-cavity device. The periodic pattern in the corrugated-cavity device resulted in a minor shoulder peak. However, it had a slight effect on the light emitted at the target wavelength because the peak wavelength remained the same and the intensity of the shoulder peak was trivial. This is the effect of Bragg scattering caused by the periodic grating structure of the corrugated device [53,54]. Moreover, Figure 14b shows that the color purity improved in the planar-cavity (*x* = 0.338, *y* = 0.606) and corrugated-cavity (*x* = 0.333, *y* = 0.588) devices compared to the reference device (*x* = 0.369, *y* = 0.556). This improvement was attributed to the microcavity effect induced by the insertion of the Ag thin film. A photograph of the emission device is shown in Figure 14c.

## 4. Conclusions

In this study, a vacuum photonic crystal layer was designed and fabricated to enhance the light extraction and improve the color purity of optoelectronic devices. The periodic line pattern of the nVPCL was determined through an FDTD simulation and optimized to extract light at green wavelengths. We evaluated the effects of applying a thin film design integrating the nVPCL and electrode to OLEDs that experienced significant optical losses because of the waveguide and SPP modes. Consequently, the EQE of the designed nVPCL device improved by 21.5% compared to the reference device and by 9.4% compared to the planar-cavity device at 10,000 cd/m^2^. In addition, the current efficiency improved by 34.6% compared to that of the reference device and the full width at half maximum decreased by 27.5%, thereby demonstrating enhanced color purity. The thin film design of the electrode using the nVPCL achieved effective light extraction by incorporating periodic patterning tuned to a specific wavelength and a vacuum component with a refractive index of 1 (to maximize the refractive index difference). Further, the intensity and color purity of the light can be enhanced through the microcavity effect. The nVPCL structure can be fabricated easily under the desired conditions by leveraging the benefits of the LIL process. The application of this structure in optoelectronic devices is expected to result in efficient light extraction and enhanced color purity. The structure optimization performed using the FDTD simulation and the design of a vacuum-incorporated thin film electrode using the LIL process presented in this study are expected to provide a perspective on highly efficient device technology applicable not only to OLEDs but also to versatile optoelectronic devices.

## 5. Patents

Based on the structure and research findings presented in this paper, the patents described below were formally submitted and jointly applied with Samsung Display Co., Ltd. (Yongin, Repubilc of Korea).

Application number: 18/305,530 (US).

Filing date: 04/24/2023.

Title of invention: Light Emitting Element and Method for Manufacturing the Same.

## Figures and Tables

**Figure 1 micromachines-15-00328-f001:**
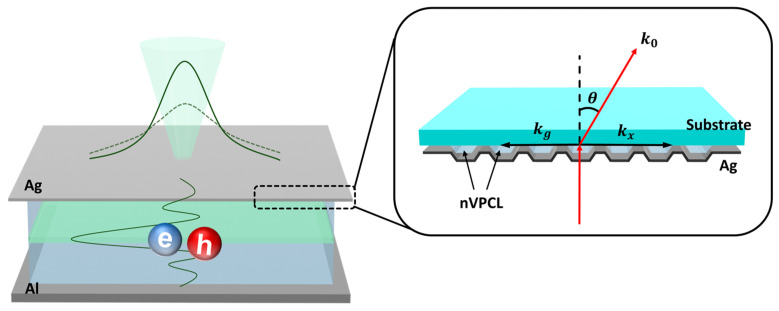
Concept of the corrugated microcavity electrode with a vacuum structure insertion.

**Figure 2 micromachines-15-00328-f002:**
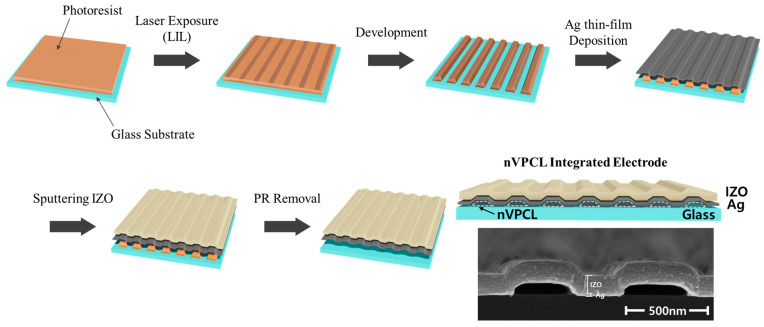
Schematic of the fabrication process of the nVPCL-integrated electrode.

**Figure 3 micromachines-15-00328-f003:**
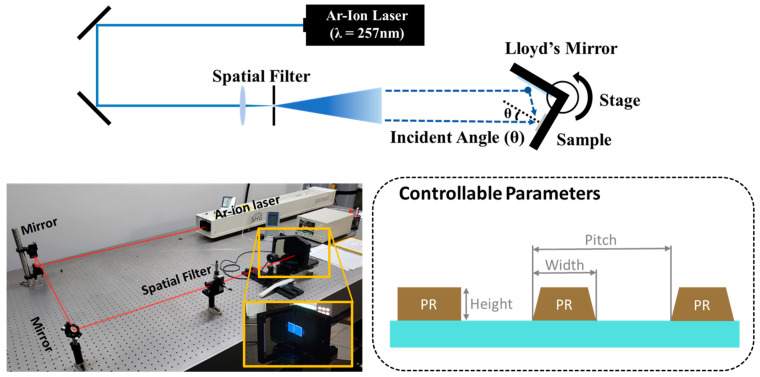
Schematic of the laser interference lithography (LIL) process.

**Figure 4 micromachines-15-00328-f004:**
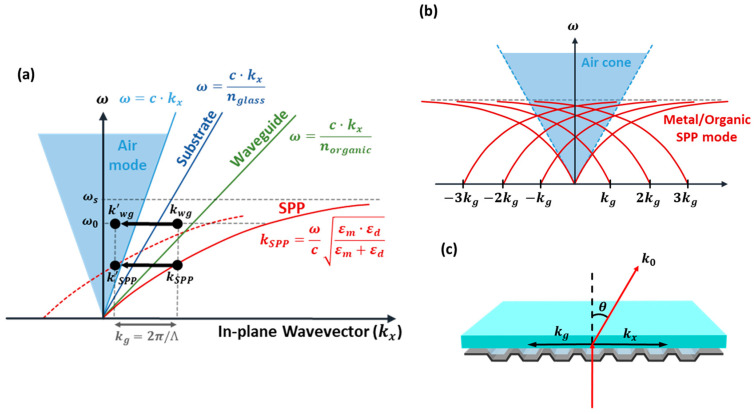
Light extraction via periodic nanostructure. (**a**) In-plane wavevector shift in the dispersion curve. (**b**) SPP mode and (**c**) waveguide mode extraction in the corrugated microcavity structure.

**Figure 5 micromachines-15-00328-f005:**
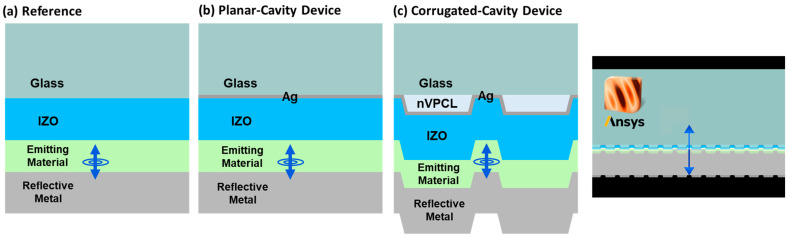
Nanophotonic computational analysis model.

**Figure 6 micromachines-15-00328-f006:**
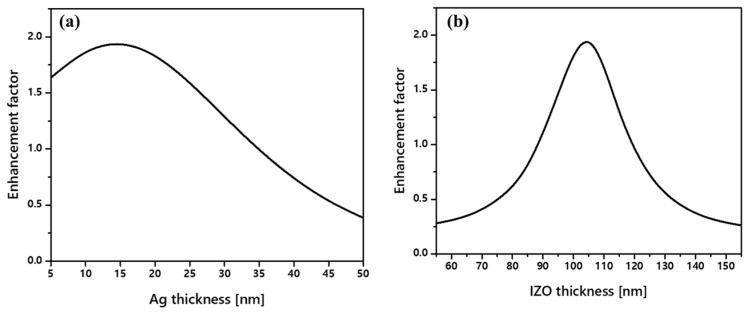
Cavity mirror thickness optimization. (**a**) Ag mirror thickness and (**b**) IZO anode thickness.

**Figure 7 micromachines-15-00328-f007:**
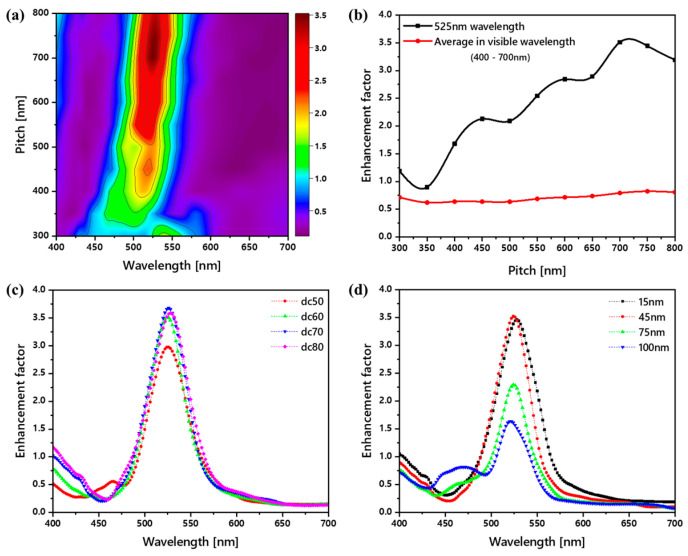
Optimization of the vacuum photonic crystal layer structure. (**a**) Contour plot illustrating the enhancement factor through a pitch sweep based on wavelength compared to the reference device. (**b**) Optimization of the pitch at the 525 nm green wavelength. Enhancements in the light extraction factor with the (**c**) width and (**d**) height sweep.

**Figure 8 micromachines-15-00328-f008:**
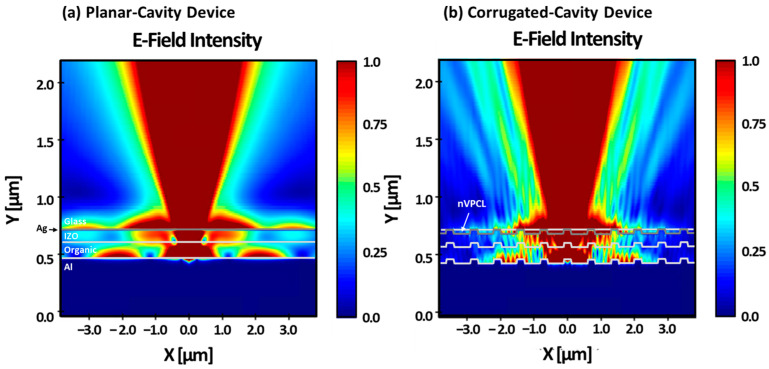
TM mode electric field distributions of (**a**) planar- and (**b**) corrugated-cavity devices.

**Figure 9 micromachines-15-00328-f009:**
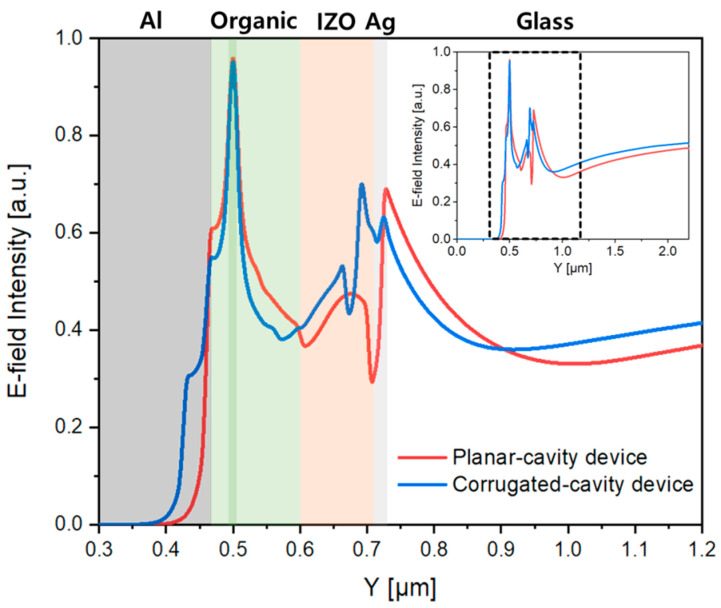
TM mode electric field line spectra of the planar- and corrugated-cavity devices.

**Figure 10 micromachines-15-00328-f010:**
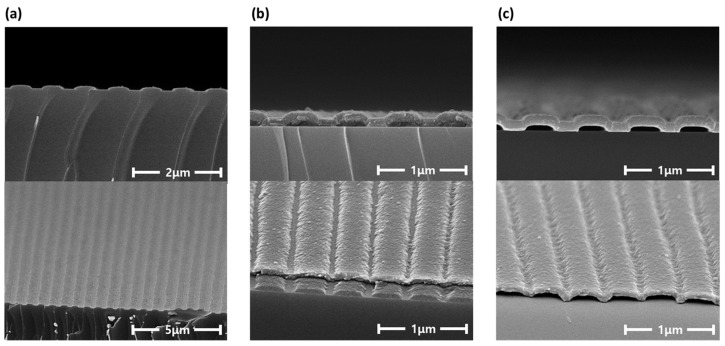
SEM images of the fabrication process of the nVPCL-integrated electrode: (**a**) line patterned photoresist, (**b**) after Ag/IZO deposition, and (**c**) after photoresist removal.

**Figure 11 micromachines-15-00328-f011:**
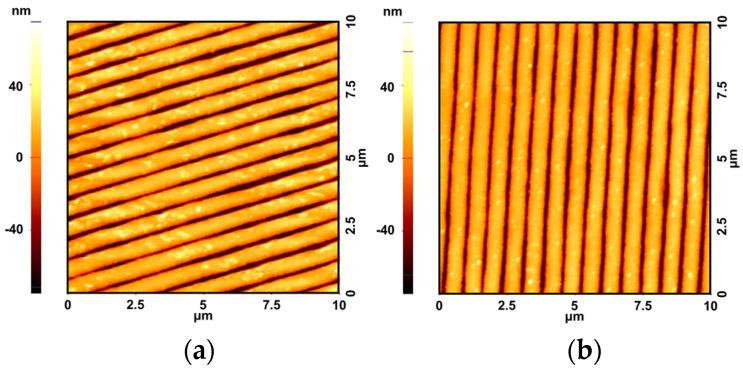
AFM images of the (**a**) corrugated Ag and (**b**) corrugated Ag/IZO structures.

**Figure 12 micromachines-15-00328-f012:**
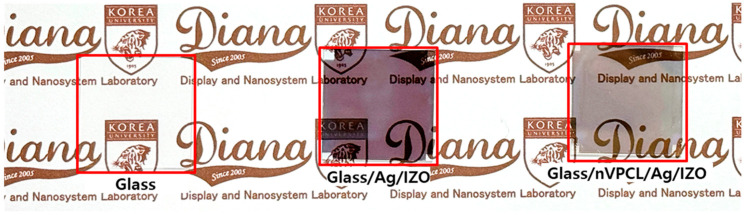
Photographs of the nVPCL-integrated electrode (top view).

**Figure 13 micromachines-15-00328-f013:**
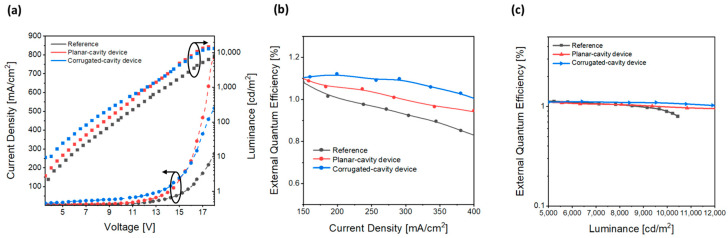
EL characteristics of the nVPCL-integrated device. (**a**) J-V-L, (**b**) J-EQE, and (**c**) L-EQE.

**Figure 14 micromachines-15-00328-f014:**
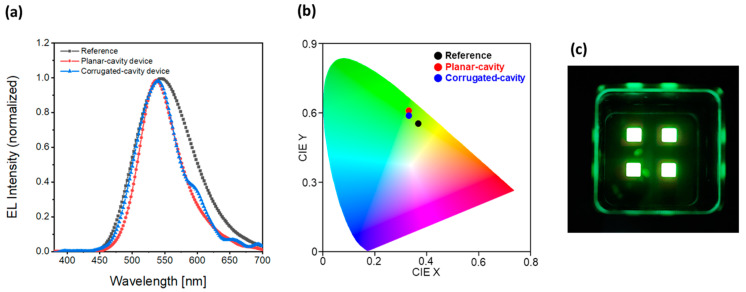
(**a**) EL intensity spectrum and (**b**) CIE 1931 color coordinate of the fabricated device. (**c**) Photograph of the operating device.

**Table 1 micromachines-15-00328-t001:** Roughness of (**a**) nVPCL/Ag, (**b**) nVPCL/Ag/IZO, (**c**) Glass/Ag, and (**d**) Glass/Ag/IZO.

Sample	Glass/nVPCL/Ag	Glass/nVPCL/Ag/IZO	Glass/Ag	Glass/Ag/IZO
Rq [nm]	27.484	24.102 (−12.31%) ^(a)^	3.058	2.318 (−24.20%) ^(a)^
Ra [nm]	22.253	19.830 (−10.89%) ^(a)^	2.379	1.781 (−25.14%) ^(a)^

^(a)^ Reduction ratio of roughness via IZO insertion.

## Data Availability

The data presented in this study are available upon reasonable request from the corresponding author.

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
