# Peer review of "Hollow Microcavity Electrode for Enhancing Light Extraction"

_micromachines, 2024, doi:10.3390/mi15030328_

Round 1
Reviewer 1 Report
Comments and Suggestions for Authors
The authors discuss a very important research topic regarding improving the operation of, among others, organic diodes. The article definitely fits the theme of the magazine and will be interesting for readers. Despite a well-organized research methodology and good presentation of results, the authors still need to develop several topics before acceptance for publication:
· 1. How was the thickness of the deposited layers controlled and examined?
· 2. Did the layers deposited by magnetron sputtering partially cover the previously prepared grooves and did this have a negative impact on the final structure?
· 3. In AFM microscopy, there were no measurements of the substrate with only a groove without the Ag and IZO layer.
· 4. The authors should explain why they won IZO and not AZO, FTO or ITO. There was also no development of this topic in the introduction.
· 5. Table 1 should be corrected.
· 6. The F20 device allows you to determine n and k only if you know the exact thickness of the layers and on a flat surface.
After developing these threads and taking into account the comments, the article should be considered for publication again.
Author Response
Dear Reviewer
Thank you for your constructive feedback on our paper. We have compiled our response in an attached document.
Sincerely yours,
Byeong-Kwon Ju

Reviewer 2 Report
Comments and Suggestions for Authors
The authors present their research on an innovative hollow micro-cavity electrode design intended for potential applications in Organic Light-Emitting Diode (OLED) technology, demonstrating a notable capacity to enhance light extraction efficiency. The prototype is beautifully demonstrated, and the device's performance is comprehensively characterized using a variety of methodologies. Parameters are optimized through Finite-Difference Time-Domain (FDTD) simulations. The manuscript aligns with the journal's scope, leading the reviewer to suggest reconsideration for publication in Micromachines, contingent upon the authors addressing specific concerns raised during the evaluation process:
1. In its current form, the manuscript's structure appears somewhat redundant. A recommended restructuring involves consolidating the simulation and fabrication details of the electrode and device under Section 2 (Methodologies), while dedicating Section 3 (Results and Discussion) exclusively to the characterization of the electrode and device. This modification aims to enhance the overall coherence and organization of the manuscript. This is my only major comment, the following comments are just minor issues.
2. Following comment 1, try to put the simulation (optimization) of the parameters in front of the fabrication part.
3. Page 2/18 line 50, ‘… thin films made of different materials’, which materials? Consider naming a few as examples.
4. Page 2/18 line 57, ‘… organic layers because of interactions with surface plasmon polaritons (SPPs)’. This sentence is a little confusing, do you mean ‘because of surface plasmon polaritons (SPPs)’.
5. The authors utilized the Laser Interference Lithography (LIL) method for patterning, emphasizing its capability to offer a considerable exposure area. However, the specific dimensions of this exposure area, particularly in terms of wafer-scale compatibility, remain undisclosed. It would be beneficial to provide details on the size and compatibility for a comprehensive understanding. Moreover, a comparative analysis with the Displacement Talbot Lithography (DTL) method (doi: 10.1016/j.mssp.2023.107311) regarding its merits and drawbacks could provide valuable insights into the selection rationale.
6. Consider removing Fig 2, since Fig. 3a already contains all the information in Fig 2.
7. In Fig 5, the authors listed height, pitch, width and taper as controllable parameters during the LIL process. How is the taper adjusted in LIL, (particularly, with your setup in Fig. 5). Please add some descriptions.
8. Increase the font size of the inset in Fig 10.
9. Upon examination of the cross-sectional Scanning Electron Microscopy (SEM) images in Fig. 11, it is evident that all the cavities exhibit rounded corners. In contrast, the simulation model (as depicted in Fig. 6) has cavities with sharp corners. This prompts the question of how significantly, if at all, this discrepancy influences the ultimate simulation results.
10. In Fig 15a, anomaly is observed in the EL intensity curve of the corrugated-cavity case within the wavelength range approximately from 580nm to 700nm. What is the cause of this anomaly?
Comments on the Quality of English LanguageThe English language is fine but it needs to be checked thoroughly before resubmission.
Author Response

(The authors gave the same response as above.)
